# The Role of Native T1 and T2 Mapping Times in Identifying PD-L1 Expression and the Histological Subtype of NSCLCs

**DOI:** 10.3390/cancers15123252

**Published:** 2023-06-20

**Authors:** Chandra Bortolotto, Gaia Messana, Antonio Lo Tito, Giulia Maria Stella, Alessandra Pinto, Chiara Podrecca, Riccardo Bellazzi, Alessia Gerbasi, Francesco Agustoni, Fei Han, Marcel Dominik Nickel, Domenico Zacà, Andrea Riccardo Filippi, Olivia Maria Bottinelli, Lorenzo Preda

**Affiliations:** 1Diagnostic Imaging and Radiotherapy Unit, Department of Clinical, Surgical, Diagnostic, and Pediatric Sciences, University of Pavia, 27100 Pavia, Italy; chandra.bortolotto@unipv.it (C.B.); antonio.lotito01@universitadipavia.it (A.L.T.); alessandra.pinto01@universitadipavia.it (A.P.); andrea.filippi@unipv.it (A.R.F.); oliviamaria.bottinelli@unipv.it (O.M.B.); 2Radiology Institute, Fondazione IRCCS Policlinico San Matteo, 27100 Pavia, Italy; 3Unit of Respiratory Diseases, Department of Medical Sciences and Infective Diseases, Fondazione IRCCS Policlinico San Matteo, 27100 Pavia, Italy; g.stella@smatteo.pv.it; 4Department of Internal Medicine and Medical Therapeutics, University of Pavia, 27100 Pavia, Italy; 5Department of Electrical, Computer and Biomedical Engineering, University of Pavia, 27100 Pavia, Italy; chiara.podrecca01@universitadipavia.it (C.P.); riccardo.bellazzi@unipv.it (R.B.); alessia.gerbasi01@universitadipavia.it (A.G.); 6Department of Medical Oncology, Fondazione IRCCS Policlinico San Matteo, 27100 Pavia, Italy; f.agustoni@smatteo.pv.it; 7MR Application Predevelopment, Siemens Healthcare GmbH, Allee am Roethelheimpark 2, 91052 Erlangen, Germany; fei.han@siemens-healthineers.com (F.H.); marcel.nickel@siemens-healthineers.com (M.D.N.); 8Siemens Healthcare, 20128 Milano, Italy; domenico.zaca@siemens-healthineers.com; 9Department of Radiation Oncology, Fondazione IRCCS Policlinico San Matteo, 27100 Pavia, Italy

**Keywords:** non-small cell lung cancer (NSCLC), programmed death-ligand 1 (PD-L1), lung MRI, T1 mapping, T2 mapping

## Abstract

**Simple Summary:**

T1 and T2 mapping are MRI techniques that are routinely used in the evaluation of both benign and malignant lesions in different organs and tissues. Immunohistochemical analysis is the gold standard method to assess the programmed death-ligand 1 protein (PD-L1) expression status in patients diagnosed with non-small cell lung cancer (NSCLC) to guide immunotherapy. There have been no studies on the correlation between T1 and T2 mapping values and PD-L1 expression of NSCLCs nor on the correlation between T2 mapping values and histological subtypes of lung tumors. Therefore, we present our preliminary results on the evaluation of the possible association of T1 and T2 mapping values with PD-L1 TPS and of their potential in distinguishing between the different histological subtypes of NSCLCs. In the future, T1 values could offer the possibility to help in the diagnosis and pathological classification of NSCLC through a non-invasive MRI exam.

**Abstract:**

We investigated the association of T1/T2 mapping values with programmed death-ligand 1 protein (PD-L1) expression in lung cancer and their potential in distinguishing between different histological subtypes of non-small cell lung cancers (NSCLCs). Thirty-five patients diagnosed with stage III NSCLC from April 2021 to December 2022 were included. Conventional MRI sequences were acquired with a 1.5 T system. Mean T1 and T2 mapping values were computed for six manually traced ROIs on different areas of the tumor. Data were analyzed through RStudio. Correlation between T1/T2 mapping values and PD-L1 expression was studied with a Wilcoxon–Mann–Whitney test. A Kruskal–Wallis test with a post-hoc Dunn test was used to study the correlation between T1/T2 mapping values and the histological subtypes: squamocellular carcinoma (SCC), adenocarcinoma (ADK), and poorly differentiated NSCLC (PD). There was no statistically significant correlation between T1/T2 mapping values and PD-L1 expression in NSCLC. We found statistically significant differences in T1 mapping values between ADK and SCC for the periphery ROI (*p*-value 0.004), the core ROI (*p*-value 0.01), and the whole tumor ROI (*p*-value 0.02). No differences were found concerning the PD NSCLCs.

## 1. Introduction

Lung cancer has the highest mortality rate among all cancers [1], with a 5-year survival of about 17.8% [2]. The World Health Organization (WHO) categorizes lung tumors into two groups: small cell lung cancer (SCLC) and non-small cell lung cancer (NSCLC), with the latter covering 85% of cases [3]. NSCLC is further sub-categorized into lung adenocarcinoma and lung squamous cell carcinoma [4].

Patients with NSCLC are often diagnosed at advanced stages [5], with cough and dyspnea as common symptoms. In this group of patients, for whom surgery is not indicated, the availability of predictive biomarkers for target molecular therapy or immunotherapy has opened new treatment possibilities in addition to conventional chemo-radiotherapy. Immunotherapy aims to modulate the immune system, reactivating it against cancer cells. To this end, with immunohistochemical analysis, the expression of the programmed death-ligand 1 protein (PD-L1) by the tumor cells is quantified. The PD-1/PD-L1 axis plays an important role in tumorigenesis and tumor development, since the binding of PD-1, expressed by lymphocytes, to its ligand PD-L1 causes downregulation of the T-cell response, directly suppressing the endogenous anti-tumor cytolytic T-cell activity. Monoclonal antibodies that block the interaction between PD-1 and PD-L1 abrogate the immune tolerance exerted by tumors through the PD-1/PD-L1 pathway.

The pathological test, either performed on tissue obtained through navigational bronchoscopy or computed tomography-guided, is the gold standard to determine the histological subtype of lung cancer and to carry out the immunohistochemical analysis in order to quantify the PD-L1 tumor proportion score (TPS). However, the biopsy is an invasive diagnostic test associated with high perioperative complication rates and severe limitations in patients with poor compliance [6]; moreover, it samples only a part of the tumor. For these reasons, in recent times, the need has emerged to find a noninvasive alternative way to identify the histologic subtype of lung tumors and quantify PD-L1 expression, providing us with information on the whole tumor rather than on a small sample of it.

The application of multiparametric magnetic resonance imaging (MRI) to lung cancer analysis is a thoroughly explored field since MRI provides better tissue characterization compared to computed tomography (CT) while involving no exposure to ionizing radiation [7,8].

T1 mapping is an MRI technique that can quantify the longitudinal relaxation time of water protons in tissues [9]. It has been used mainly in cardiovascular imaging as an important tool for the characterization of myocardial tissue [10]. Previous studies have been focused on the use of T1 mapping in pulmonary diseases, specifically on functional assessment in chronic obstructive pulmonary disease (COPD) [11,12], on the differentiation between benign and malignant lung lesions, based on the different water content [13], and on the identification of lung cancer pathological types [14] and their correlation with Ki-67 expression [15].

T2 mapping instead is able to calculate the T2 time, i.e., the transverse relaxation time of water protons, displaying it voxel-vice on a parametric map [9]. It has been applied mainly to the characterization of myocardium-related diseases [16,17], but also to many other clinical conditions, including prostate tumors [18], breast tumors [19], ovarian cancer [20], uterine lesions [21], and osteoarthritis [22].

To the best of our knowledge, there have been no studies on the correlation between T1 and T2 mapping values and PD-L1 expression of NSCLCs nor on the correlation between T2 mapping values and the histological types of lung tumors. Therefore, the objectives of our study were (1) to investigate the possible association of T1 and T2 mapping values with PD-L1 TPS and (2) to evaluate their potential in distinguishing between the different histological subtypes of NSCLCs.

## 2. Materials and Methods

### 2.1. Patients

From April 2021 to December 2022, we prospectively enrolled a consecutive group of 35 patients diagnosed with locally advanced NSCLC (according to TNM classification VIII edition, stage III A-C cancer and T ≥ 2); all of them have previously undergone complete tumor staging and pathological characterization as well as an immunohistochemistry assay for PD-L1 quantification. The following patients were excluded: (I) patients who had no adequate compliance capabilities and characteristics to guarantee the correct execution of the MRI examination; (II) those who had received treatment before the MRI; (III) patients with lung tumors not classified as NSCLC. PD-L1 TPS was quantified through the 22C3 pharmDx assay (Agilent Technologies, Carpinteria, CA) (Figure 1). Based on immunohistochemistry (IHC) results, the tumors were distinguished into two groups: no PD-L1 expression (<1%), and positive PD-L1 expression (≥1%). This study was conducted in accordance with the ethical standards laid down in the Declaration of Helsinki and was approved by the Ethics Committee of the Hospital.

### 2.2. MRI Examination

All patients underwent MRI lung examination. MRI was performed with a 1.5 T system (MAGNETOM Aera; Siemens Healthcare, Erlangen, Germany) using a 32-channel spine array. During the examination, both free-breathing and breath-hold sequences were used. Plain scan sequences included: axial and coronal volumetric interpolated breath-hold examination (VIBE) T1 sequence, axial T1 and T2 mapping, and functional sequences of ventilation. Axial T1 mapping was performed using a research application based on the Look-Locker method. Parameters included: TR 3 ms, TE 1,32 ms, slice thickness 8 mm, interval 0 mm, scan matrix 192 × 192, FOV 380 mm, number of slices 4/6, number of contrasts 16, flip angle 8°, sampling time 48 msec. For axial T2 mapping, a research application based on a radially sampled technique combined with turbo-spin-echo (TSE) multi-echo imaging was used with the following acquisition parameters: TR 1429 ms, TE 8,6 ms, slice thickness 5 mm, interval 0 mm, scan matrix 256 × 256, FOV 350 mm, number of slices 19, radial views 396.

### 2.3. Post-Processing of MRI Images

MRI images were analyzed with the Software PACS Synapse (Fujifilm medical systems, U.S.A., Inc., Lexington, MA, USA). Being T1 and T2 mapping images characterized by parametric maps reconstructed by calculating the values on a pixel-by-pixel basis, pixel intensities correspond to either T1 or T2 values. For each sequence of T1 and T2 mapping, different regions of interest (ROIs) were manually traced on all the slices where the tumor was comprised, to include the whole extent of the tumor and avoid the surrounding normal tissue, and then the mean T1 and T2 values were calculated (*whole tumor ROI*). In addition, four ROIs were manually traced on the slice where the tumor had the largest diameter: one with a thickness of 6 mm to include only the peripheral portion of the tumor (*periphery ROI*), one freehand ROI to include only the central portion of the tumor (*core ROI*) (Figure 2), two ROIs with a mean thickness of 3 mm and 6 mm, to include the surrounding lung parenchyma adjacent to the tumor (*microenvironment 3 mm ROI* and *microenvironment 6 mm ROI,* respectively). One circular ROI with a diameter of 6 mm was traced on the normal lung parenchyma, away from the tumor (*normal lung ROI*). For each of the six ROIs selected, both mean T1 and T2 mapping values were computed, thus obtaining 12 mapping features for each patient.

### 2.4. Statistical Analysis

RStudio was used to analyze the data and apply statistical methods. Two main correlation analyses have been performed: the first one between the T1/T2 mapping values and the binary target class PD-L1 expression (expressed if ≥1% or unexpressed if <1%); and the second analysis was performed between the T1/T2 mapping values and the lung histological subtype: squamocellular carcinoma (SCC), adenocarcinoma (ADK), poorly differentiated NSCLC (PD). Each ROI value attribute has been tested individually to assess significant differences in the continuous variables (the mapping values) by the categorical target class. For PD-L1 analysis, the Wilcoxon–Mann–Whitney test has been used, while for histotype analysis the Kruskal–Wallis alternative for three groups has been used, with a consequent post-hoc Dunn test. The difference between *core ROI* and *peripheral ROI* was also tested with the Wilcoxon–Mann–Whitney test, both for T1 and T2 mapping, to check if there was an actual difference in the mean mapping values between the two attributes. Nonparametric tests have been performed since there were not sufficient patients to assume hypotheses on the data distribution. Significant differences were considered when the *p*-value < 0.05. We report significant results along with the false discovery rate for multiple testing.

## 3. Results

### 3.1. Patients’ Clinical Data

A total of 35 patients were included in the study. Among them, 26 (74%) were males and 9 (26%) were females. All the patients were of Caucasian ethnicity, aged between 49 and 84, with an average age of 68. Excluding five patients for whom PD-L1 expression could not be determined, 18 patients (60%) had a positive PD-L1 expression (≥1%) and 12 (40%) had no PD-L1 expression (<1%). Regarding the histological subtype, among the 35 patients, 2 of them were excluded because they were missing a defined histological type (NSCLC not otherwise specified, NOS); there were 11 patients (33%) with lung SCC, 13 patients (40%) with lung ADK, and 9 patients (27%) with PD NSCLC (Table 1). The mean interindividual values in the patients are shown in Table 2 and Table 3. Table 4 shows the amount of missing values for each target class and feature.

### 3.2. Correlation between T1 and T2 Mapping Values and PD-L1 Expression in NSCLC

From the univariate analyses performed on the mapping features and the PD-L1 expression with the Wilcoxon–Mann–Whitney test, there were no statistically significant differences between the two groups.

### 3.3. Correlation between T1 and T2 Mapping Values and Histological Subtype of NSCLC

Firstly, with the Kruskal–Wallis test we checked if there were any differences at all in the three histological subtype groups. The result (Table 5) was that T1 mapping in the *periphery ROI* had a *p*-value of 0.013 (with false discovery rate, FDR = 0.12), and that T1 mapping in the *core ROI* had a *p*-value of 0.046 (with FDR = 0.18). The boxplots for the T1/T2 values are shown in Figure 3, Figure 4 and Figure 5. In order to determine which groups were different from the others in the T1/T2 values found significant with the Kruskal–Wallis test, post-hoc paired testing was conducted. The Dunn test with Bonferroni adjustment procedure was performed. For the *T1 periphery ROI*, the ADK group was significantly different from the SCC group (*p*-value 0.004, adjusted 0.01), while there was not a statistically significant difference between ADK and PD group (*p*-value 0.14, adjusted 0.40) as well as between the SCC and PD groups (*p*-value 0.26, adjusted 0.78). In addition, for the *T1 core ROI*, the ADK group was significantly different from the SCC group (*p*-value 0.01, adjusted 0.04), while there was not a statistically significant difference between the ADK and PD group (*p*-value 0.41, adjusted 1), as well as between the SCC and PD groups (*p*-value 0.15, adjusted 0.46). Overall, the *T1 whole* ROI gave a *p*-value of 0.063 in the Kruskal–Wallis test, with a FDR = 0.17, and the Dunn test difference between the ADK group and the SCC group had a *p*-value of 0.02 (adjusted 0.06), while there were no statistically significant differences between the ADK and PD groups (*p*-value 0.15, adjusted 0.46) and between the SCC and PD groups (*p*-value 0.40, adjusted 1) (Table 6). The mean T1 and T2 mapping values between the *core ROI* and the *periphery ROI* are not statistically different, and they are not statistically different from the whole tumor ROI.

In Appendix A are listed all the T1 and T2 mapping times for each ROI for each patient, as well as PD-L1 TPS.

## 4. Discussion

In general clinical practice, the histological subtype of lung tumors is evaluated through pathological analysis, while the expression of PD-L1 by the tumor cells is quantified with immunohistochemical analysis. However, a biopsy of the tumor is necessary, which is an invasive diagnostic test that only samples a part of the tumor and carries important limitations in patients with poor compliance. For these reasons, in recent times there have been many attempts to find a noninvasive alternative way to the bioptic test. Magnetic Resonance Imaging (MRI) can provide not only morphologic information but also better tissue characterization with respect to Computed Tomography (CT).

Our results suggest that native T1 mapping times estimated during MRI of the thorax could be used to noninvasively identify the histological subtype of NSCLCs. Furthermore, we demonstrated that T1 and T2 mapping values seem unable to distinguish between tumors with positive PD-L1 expression and those without PD-L1 expression.

We used a protocol to study the lungs with MRI that included a complete morphological assessment and T1 and T2 mapping sequences, followed by the quantification of T1 and T2 times of lung tumors.

T1 mapping is defined by the pixel-to-pixel illustration of absolute T1 relaxation times. Elevated T1 relaxation times have been associated with increased extracellular compartment volume; thus, conditions such as protein deposition or fibrosis can be detected using native T1 mapping [23,24,25,26]. Yang et al. demonstrated that T1 mapping could distinguish benign from malignant lung lesions due to their different water content; however, the range of T1 values for malignant lung lesions was broad, and the reason could lie in the fact that it varies according to the different pathological types of lung cancer [13]. Recently, many studies have been focused on T1 mapping as an independent imaging biomarker for characterizing histological lung cancer type. Li et al. have reported a significant difference in T1 values between SCLC and ADK, and between SCC and ADK; specifically, they found higher T1 values in SCC and SCLC compared to ADK. Additionally, they demonstrated a statistically significant difference in apparent diffusion coefficient (ADC) and T1 values between the moderately and highly differentiated group and the poorly differentiated group of lung tumors [15]. Jiang et al. observed significantly higher T1 values in SCLC compared to ADK and SCC; however, there was no significant difference between ADK and SCC [14]. In our study, for the *periphery ROI*, *core ROI* and *whole tumor ROI*, T1 values of ADKs were significantly higher with respect to those of SCCs. This could be related to the different extracellular matrix (ECM) composition of the two histological subtypes of tumors, with the ADKs included in our study being more fibrotic than the SCCs. Indeed, due to their different anatomical locations, adenocarcinoma and squamous cancer cells are exposed to different ECM, which is an important regulator of cell behavior in multiple cancer types [27]. Primary lung tumors, both the ADK and SCC subtypes, have increased and altered fibrillar collagen deposition, consistent with a fibrotic response. On one hand, tenascin-C, a glycoprotein that, if activated, promotes collagen deposition, is significantly upregulated in fibrotic lungs and in lung ADK [28]; on the other hand, the glycoprotein periostin, expressed by activated fibroblasts, is higher expressed in SCCs compared to ADKs [29].

Since the association between ECM expression and prognosis has been documented in general in NSCLCs [30], T1 mapping values could also be a surrogate for ECM quantification to correlate with prognosis, but further studies are necessary.

T2 relaxation time represents the time constant governing the exponential decay of transverse magnetization. The fractional increment in T2 is larger than the one in T1 when the water content is increased, making T2 mapping a tool for detecting edema [17]. In our study, T2 values did not show significant differences between histological subtypes of NSCLC; however, we reported the T2 mean values of the different subtypes in order to have reference values for future research.

We selected five different ROIs for estimating T1 and T2 values in order to evaluate potential differences in tissue composition among different regions of the tumor, i.e., the whole extent of the tumor, its peripheral portion avoiding the possible necrotic center, its central portion, and the lung tissue strictly adjacent to the tumor. However, there was not a statistically significant difference between the different tumor regions.

Furthermore, no statistically significant difference was identified between PD-L1 tumor proportion score (TPS) neither with T1 nor with T2 mapping values, potentially highlighting the fact that there were no differences in ECM composition or cellular inflammatory infiltrates detectable with MR between the two groups, even considering different regions of the tumors; further studies on the cellular microenvironment of specific lung tumors’ subtypes are needed to confirm our results, as well as the development of new and more advanced MR sequences with better sensitivity to small modifications of imaging parameters.

Tumor heterogeneity is a challenge of modern oncology since it can have a negative impact on clinical outcomes. Yet, the assessment of heterogeneity on the histological sample following biopsy may be tainted by a high level of sampling bias. T1 mapping may be a promising MRI sequence for evaluating the intrinsic properties of tumors, also in association with other MRI sequences. Firstly, MRI can be repeated regularly, also under therapy, without the use of ionizing radiation; furthermore, the T1 mapping sequence we used can be performed in short scanning times and, thus, it can be easily incorporated into MR exams.

Our study was the first research project with the aim to investigate the relationship between PD-L1 expression and T1 or T2 mapping values, however, it presents several limitations: in the first place, we used a set of measurements which were collected by a single trained radiologist, so the independent work of two or more radiologists could have been more useful in evaluating the presence or absence of inter-observer agreement and further increasing the strength of our data. Moreover, low proton density, susceptibility artifacts, and motion artifacts may affect the reproducibility of native T1 mapping in lung lesions.

Another factor to be considered is the heterogeneity both within the neoplastic tissue and at the interface of tumor–normal lung or tumor–atelectasis, which could lead to some bias in the measurement of T1 and T2 values; the ROI areas have been traced on tumors of different sizes and shapes, so they may have included smaller or larger necrotic regions that could have affected our measurements. Finally, this is a monocentric study with a small sample size.

Further studies are necessary to validate T1 mapping values in determining the histological subtype of NSCLCs and to investigate the potential correlation of T1 and T2 values with PD-L1 TPS.

## 5. Conclusions

In conclusion, our results suggest that MRI T1 longitudinal relaxation times may detect differences among the histological subtypes of NSCLCs, reflecting their tissue characteristics, while T1 and T2 mapping are not efficient in distinguishing between NSCLCs with different PD-L1 expression statuses.

## Figures and Tables

**Figure 1 cancers-15-03252-f001:**
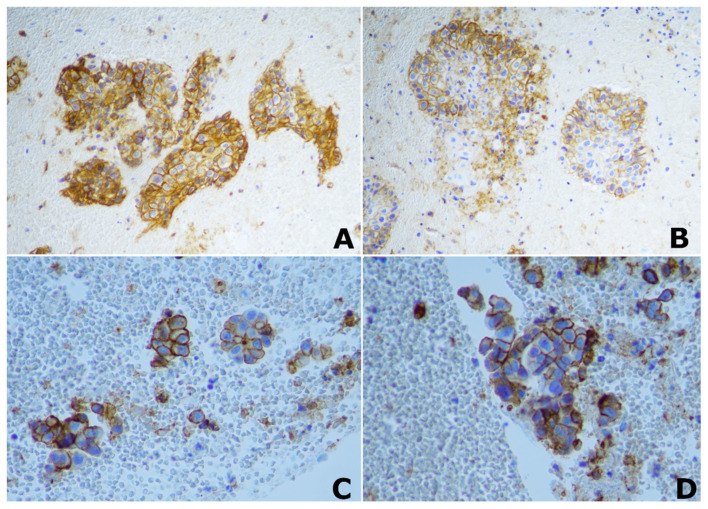
Immunohistochemical (IHC) staining of PD-L1. ×20 magnification image of positive PD-L1 IHC staining in two different patients with lung squamocellular carcinoma (**A**,**B**); ×40 magnification image of positive PD-L1 IHC staining in two different patients with lung adenocarcinoma (**C**,**D**).

**Figure 2 cancers-15-03252-f002:**
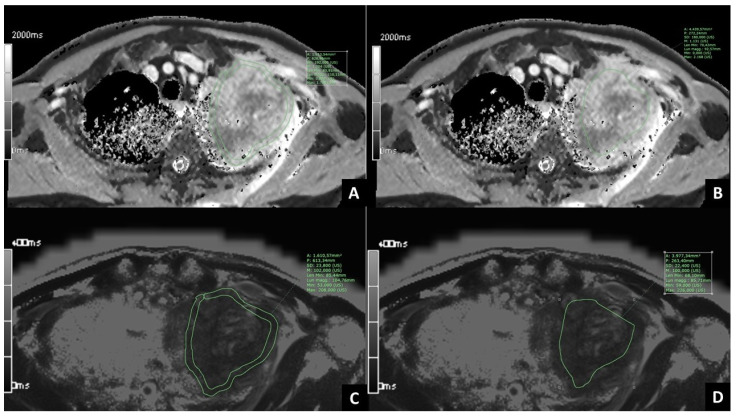
Examples of manually traced ROIs in a patient with lung cancer of the left upper lobe. (**A**,**B**). T1 mapping images with *periphery ROI* (**A**) and *core ROI* (**B**) traced on the slice comprising the largest diameter of the tumor. (**C**,**D**). T2 mapping images with *periphery ROI* (**C**) and *core ROI* (**D**) traced on the slice comprising the largest diameter of the tumor.

**Figure 3 cancers-15-03252-f003:**
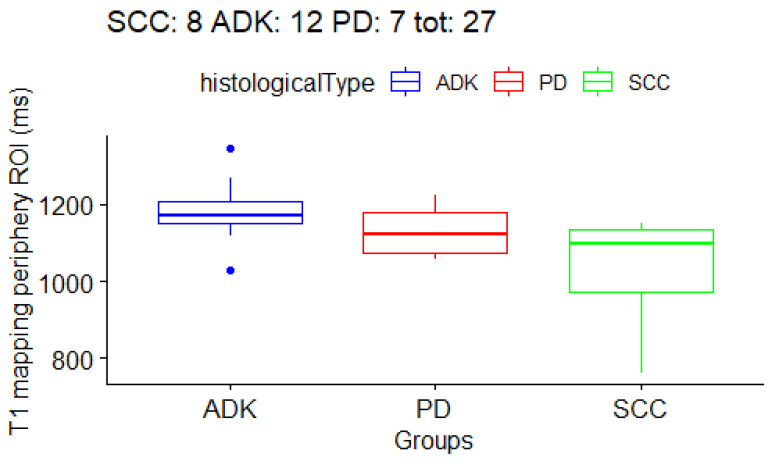
Boxplot for *T1 mapping periphery ROIs*. One boxplot for each histological subtype (SCC, ADK, PD). Distribution of 27 patients.

**Figure 4 cancers-15-03252-f004:**
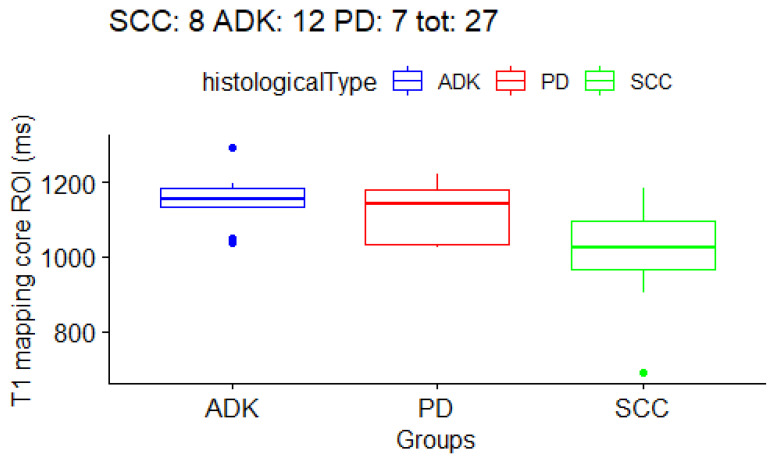
Boxplot for *T1 mapping core ROIs*. One boxplot for each histological subtype (SCC, ADK, PD). Distribution of 27 patients.

**Figure 5 cancers-15-03252-f005:**
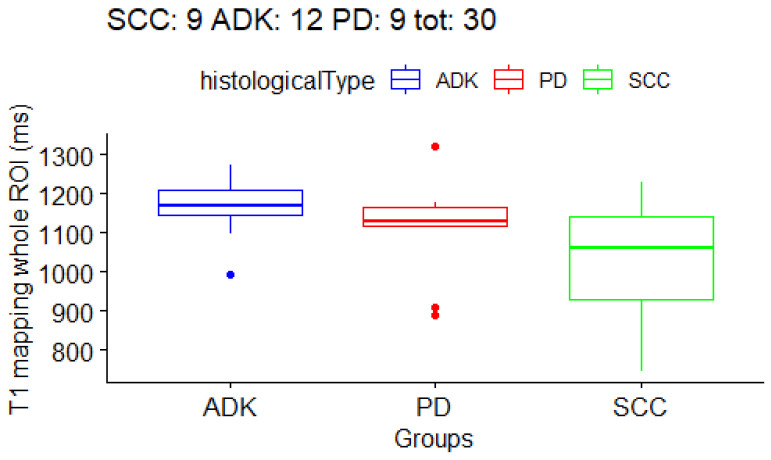
Boxplot for *T1 mapping whole tumor ROIs*. One boxplot for each histological subtype (SCC, ADK, PD). Distribution of 30 patients.

**Table 1 cancers-15-03252-t001:** Main patients’ characteristics.

Main Patients’ Characteristics	
Characteristic	Value
**Age (n = 35)**	
Median	68
Range	49–84
**Sex (n = 35)**	
Females	26% (9)
Males	74% (26)
**NSCLC histotype (n = 33)**	
Adenocarcinoma	40% (13)
Squamocellular carcinoma	33% (11)
Poorly-differentiated NSCLC	27% (9)
**PD-L1 expression (n = 30)**	
PD-L1 ≥ 1	60% (18)
PD-L1 < 1	40% (12)

**Table 2 cancers-15-03252-t002:** T1 and T2 mapping data distribution on each of the six selected ROIs, for the two different groups of PD-L1 expression.

	μ ± σ T1 Mapping		μ ± σ T2 Mapping	
	PD-L1 < 1%	PD-L1 ≥ 1%	PD-L1 < 1%	PD-L1 ≥ 1%
whole tumor	1109 ± 120 ms	1108.2 ± 150 ms	108.7 ± 28 ms	105.6 ± 28 ms
periphery	1150.5 ± 102 ms	1100.1 ± 133 ms	104.9 ± 24 ms	106 ± 31 ms
core	1106.8 ± 73 ms	1099.7 ± 148 ms	96.6 ± 24 ms	105 ± 31 ms
microenv. 3 mm	1014.1 ± 145 ms	934.9 ± 73 ms	180.4 ± 47 ms	180 ± 49 ms
microenv. 6 mm	938.9 ± 77 ms	912.9 ± 81 ms	189.6 ± 48 ms	186.6 ± 51 ms
normal lung	897.9 ± 51 ms	856.1 ± 83 ms	237.4 ± 25 ms	231.7 ± 27 ms

**Table 3 cancers-15-03252-t003:** T1 and T2 mapping data distribution on each of the six selected ROIs, for the three different histological subtype groups.

	μ ± σ T1 Mapping		μ ± σ T2 Mapping	
	SCC	ADK	PD	SCC
whole tumor	1032.9 ± 154 ms	1169.2 ± 76 ms	1107.1 ± 133 ms	101 ± 27 ms
periphery	1029 ± 150 ms	1180.5 ± 79 ms	1129.1 ± 68 ms	103.5 ± 31 ms
core	1005.3 ± 156 ms	1153.4 ± 68 ms	1115.9 ± 84 ms	100.6 ± 28 ms
micro env. 3 mm	928.3 ± 50 ms	978.3 ± 92 ms	1020.5 ± 188 ms	175.8 ± 57 ms
micro env. 6 mm	905.8 ± 61 ms	955.9 ± 73 ms	922 ± 130 ms	179.3 ± 57 ms
normal lung	885 ± 70 ms	877.8 ± 71 ms	865.7 ± 87 ms	233.9 ± 22 ms

**Table 4 cancers-15-03252-t004:** Missing values percentage for each feature, considering both histotype and PD-L1 target class.

Missing Values	Histotype: 2/35 (5%) → Tot 33	PD-L1: 5/35 (14%) → Tot 30
T1 whole tumor	3/33 (9%)	3/30 (10%)
T1 periphery	6/33 (18%)	5/30 (17%)
T1 core	6/33 (18%)	5/30 (17%)
T1 micro env. 3 mm	7/33 (21%)	6/30 (20%)
T1 micro env. 6 mm	7/33 (21%)	6/30 (20%)
T1 normal lung	6/33 (18%)	5/30 (17%)
T2 whole	1/33 (3%)	1/30 (3%)
T2 periphery	2/33 (6%)	1/30 (3%)
T2 core	2/33 (6%)	1/30 (3%)
T2 micro env. 3 mm	3/33 (9%)	2/30 (7%)
T2 micro env. 6 mm	3/33 (9%)	2/30 (7%)
T2 normal lung	2/33 (6%)	1/30 (3%)

**Table 5 cancers-15-03252-t005:** Results for histological subtype analysis, with Kruskal–Wallis *p*-values along with false discovery rate (FDR). Values in bold are for significant *p*-values.

ROI	*p*-Value	FDR
T1 whole tumor	0.063	0.176
T1 periphery	0.013	0.116
T1 core	0.046	0.176
T1 micro env. 3 mm	0.297	0.612
T1 micro env. 6 mm	0.314	0.612
T1 normal lung	0.946	0.946
T2 whole	0.350	0.612
T2 periphery	0.690	0.878
T2 core	0.937	0.946
T2 micro env. 3 mm	0.558	0.781
T2 micro env. 6 mm	0.555	0.781
T2 normal lung	0.871	0.946

**Table 6 cancers-15-03252-t006:** Results of *T1 periphery*, *T1 core* and *T1 whole tumor ROIs* for histological subtypes analysis from Kruskal–Wallis test, along with paired post-hoc analysis with the Dunn test *p*-values. Statistically significant *p*-values in bold.

	SCC	ADK	PD	*p*-Value
T1 periphery	1029 ± 150 ms	1180.5 ± 79 ms	1129.1 ± 68 ms	0.01
ADK vs. SCC	-	-	-	**0.004**
ADK vs. PD	-	-	-	0.13
SCC vs. PD	-	-	-	0.26
T1 core	1005.3 ± 156 ms	1153.4 ± 68 ms	1115.9 ± 84 ms	0.04
ADK vs. SCC	-	-	-	**0.01**
ADK vs. PD	-	-	-	0.41
SCC vs. PD	-	-	-	0.15
T1 whole	1032.9 ± 154 ms	1169.2 ± 76 ms	1107.1 ± 133 ms	0.06
ADK vs. SCC	-	-	-	**0.02**
ADK vs. PD	-	-	-	0.15
SCC vs. PD	-	-	-	0.40

## Data Availability

The data presented in this study are available in this article.

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
