# Peer review of "The Role of Native T1 and T2 Mapping Times in Identifying PD-L1 Expression and the Histological Subtype of NSCLCs"

_cancers, 2023, doi:10.3390/cancers15123252_

Round 1
Reviewer 1 Report
A few things that may help this statements in this paper become more convincing :
1. Justification of motivation of using MRI results to correlate with PD-L1 expression and different cancer subtypes as these are conventionally only be examined through higher resolution assays, for example, histopathology.
2. The post-hoc paired testing after the Kruskal Wallis test should have either a Bonferroni or Dunn-Sidak correction to the significances obtained from the series of Mann-Whitney test results. It was not clear from the description whether this correction has been added or not.
3. The raw readout T1/T2 and PD-L1 dataset should be published for the readers to check the soundness of the conclusions from the paper.
Author Response
Dear Reviewer,
We appreciate the constructive suggestions and the attentive comments you have made. We have incorporated them in the Article; changes are marked-up within the file “Revised Manuscript with Track Changes”.
Our response follows.
- Justification of motivation of using MRI results to correlate with PD-L1 expression and different cancer subtypes as these are conventionally only be examined through higher resolution assays, for example, histopathology.
Thank you for your comment. We added a small paragraph at the beginning of the discussion, in addition to the one already present in the introduction regarding this topic.
- The post-hoc paired testing after the Kruskal Wallis test should have either a Bonferroni or Dunn-Sidak correction to the significances obtained from the series of Mann-Whitney test results. It was not clear from the description whether this correction has been added or not.
Thank you for suggesting the most appropriate post-hoc test for histotype analysis. To summarize, we used the Wilcoxon-Mann-Whitney test with false discovery rate (FDR) for the PD-L1 analyses, and we used it also to test differences in the data distributions between core and periphery ROIs, as well as from whole ROIs (both for T1 and T2 mapping). We performed Kruskal Wallis with FDR for the histotype analysis, and then changed the post hoc Wilcoxon-Mann-Whitney with the Dunn test with Bonferroni correction as you suggested, reporting p-values along with their adjustments.
- The raw readout T1/T2 and PD-L1 dataset should be published for the readers to check the soundness of the conclusions from the paper.
We included a table with the raw readout data in the Supplementary Materials.
Reviewer 2 Report
In the manuscript Bortolotto et al. described the role of native T1 and T2 mapping test in identifying the expression of PD-L1 and the histological subtype of NSCLC.
The paper is well written and flows logically. For the first time, the correlation between T1/T2 mapping and PD-L1 expression in NSCLC and the T2 mapping and histological types of lung cancer is reported.
Some comments:
-please, add more details regarding the procedure and the analysis/quantification
-a table with the patients’clinical data will be more useful
-please, provide pictures of PD-L1 staining
minor editing
Author Response
Dear Reviewer,
We appreciate the constructive suggestions and the attentive comments you have made. We have incorporated them in the Article; changes are marked-up within the file “Revised Manuscript with Track Changes”.
Our response follows.
-please, add more details regarding the procedure and the analysis/quantification
We incorporated some details regarding the post-processing of MRI images and the statistical analysis.
-a table with the patients’ clinical data will be more useful
We added a table with the main patients’ clinical data (Table 1).
-please, provide pictures of PD-L1 staining
We added Figure 1, in which is illustrated positive PD-L1 immunohistochemical staining in two patients with lung SCC and two patients with lung ADK.
Round 2
Reviewer 1 Report
Concerns has been properly addressed.